# Effect of Dewaxed Coffee on Gastroesophageal Symptoms in Patients with GERD: A Randomized Pilot Study

**DOI:** 10.3390/nu14122510

**Published:** 2022-06-16

**Authors:** Barbara Polese, Luana Izzo, Nicola Mancino, Marcella Pesce, Sara Rurgo, Maria Cristina Tricarico, Sonia Lombardi, Barbara De Conno, Giovanni Sarnelli, Alberto Ritieni

**Affiliations:** 1Digestive and Nutritional Pathophysiology Unit, Department of Clinical Medicine and Surgery, University of Naples “Federico II”, Via Pansini 5, 80131 Naples, Italy; barbara.polese@gmail.com (B.P.); nicola.mancino.36@gmail.com (N.M.); macella.pesce@unina.it (M.P.); sara.rurgo@unina.it (S.R.); barbara.deconno@gmail.com (B.D.C.); giovanni.sarnelli@unina.it (G.S.); 2Food Lab, Department of Pharmacy, University of Naples “Federico II”, Via Domenico Montesano 49, 80131 Naples, Italy; sonia.lombardi@unina.it (S.L.); alberto.ritieni@unina.it (A.R.); 3Kimbo S.p.A., Via Gian Lorenzo Bernini 20, 80129 Naples, Italy; mariacristina.tricarico@kimbo.it; 4United Nations Educational, Scientific and Cultural Organization Chair on Health Education and Sustainable Development, University of Naples “Federico II”, 80131 Naples, Italy

**Keywords:** dewaxed coffee, gastroesophageal reflux disease, C-5-HT, UHPLC Q-Orbitrap HRMS, chlorogenic acids

## Abstract

Gastroesophageal Reflux Disease (GERD) is multifactorial pathogenesis characterized by the abnormal reflux of stomach contents into the esophagus. Symptoms are worse after the ingestion of certain foods, such as coffee. Hence, a randomized pilot study conducted on 40 Italian subjects was assessed to verify the effect of standard (SC) and dewaxed coffee (DC) consumption on gastroesophageal reflux symptoms and quality of life in patients with gastrointestinal diseases. The assessment of patient diaries highlighted a significant percentage reduction of symptoms frequency when consuming DC and a significant increase in both heartburn-free and regurgitation-free days. Consequentially, patients had a significant increase of antacid-free days during the DC assumption. Moreover, the polyphenolic profile of coffee pods was ascertained through UHPLC-Q-Orbitrap HRMS analysis. Chlorogenic acids (CGAs) were the most abundant investigated compounds with a concentration level ranging between 7.316 (DC) and 6.721 mg/g (SC). Apart from CGAs, caffeine was quantified at a concentration level of 5.691 mg/g and 11.091 for DC and SC, respectively. While still preliminary, data obtained from the present pilot study provide promising evidence for the efficacy of DC consumption in patients with GERD. Therefore, this treatment might represent a feasible way to make coffee more digestible and better tolerated.

## 1. Introduction

Coffee represents one of the most popular beverages in the world. According to the latest data that was reported by the Food and Agriculture Organization of the United Nations (FAO), Italian coffee intake is around 5.6 kg per capita/year, significantly higher than overall European consumption which registers an annual per capita intake of 3.8 kg/person [1]. The chemical composition of coffee is rich in bioactive compounds, more specifically chlorogenic acid (CGA), dicaffeoylquinic acids (diCQA), feruloylquinic acids (FQA) caffeine, diterpenes, melanoidins, and trigonelline [2,3]. The evidence suggests that the occurrence of bioactive compounds is attributable to many health-promoting outcomes with reduced risk for the development of cancers, diabetes, liver, and cardiovascular disease [4,5,6]. On the other hand, coffee consumption is sometimes identified as a possible trigger for heartburn and regurgitation in Gastroesophageal Reflux Disease (GERD) patients [7,8,9].

GERD is a common chronic and relapsing condition characterized by the presence of troubling symptoms caused by the abnormal reflux of stomach contents into the esophagus [10]. The prevalence of GERD is continuously increasing, affecting one-fourth of the general population and reaching up to 27.8% in Western countries [11]. The heterogeneous presentation and the multifactorial pathogenesis of GERD make the management of GERD patients delicate and difficult [12,13,14]. Typical symptoms of GERD, heartburn and acid regurgitation, are often accompanied by a broad spectrum of atypical symptoms such as sore throat, hoarseness, cough, digestive alterations, and sleep modifications [15]. Due to their frequency and severity, GERD symptoms may often affect patients’ quality of life and daily activities [16,17,18], leading also to decreased work productivity [19]. The steady increase in GERD prevalence seems to be linked to a parallel increase in environmental risk factors such as obesity [20,21,22], tobacco smoking, decreased levels of physical activity, and the spread of incorrect dietary habits [7,23,24]. Generally, patients with GERD report a worsening of gastroesophageal reflux symptoms after ingestion of certain foods or nutrients, and this often leads to a self-managed restriction of their diet without real scientific support for this behavior [8]. Indeed, sound evidence showing a causal relationship between food consumption and GERD symptoms is still scant and controversial to date [7]. The pathophysiologic mechanism underlying the aggravation of GERD symptoms by coffee is still under debate [25,26,27]. Some of the studies reported in the literature hypothesized the capability of coffee to diminish basal lower esophageal sphincter (LES) pressure, responsible for gastroesophageal reflux and heartburn [28,29,30].

Among the least-studied molecules, the waxy elements, classified as –Nβ-Alkanoyl-5-hydroxytryptamine (C-5-HT), are naturally present in the cortical part of the coffee bean. The solubility properties of waxes (they become soluble around 65 °C) make them barely digestible and difficult to absorb for the human body [31,32]. Furthermore, they could be able to induce a mild irritation of gastric mucosa in predisposed subjects. Besides causing possible digestive problems, waxes may partially occlude taste buds, limiting the capability to savor the coffee taste. The dewaxing process is a mild innovative treatment of extraction in which the waxy layer is removed from unroasted coffee together with a small amount of caffeine with an organic solvent. Therefore, this treatment might represent a feasible way to make coffee more digestible and better tolerated by patients with GERD.

Based on the above, this study aimed to explore the relationship between standard and dewaxed coffee intake on GERD. This association is performed using a pilot study through a GERD questionnaire. The polyphenolic profile of the pods’ extract was determined through ultra-high-performance liquid chromatography coupled to a high-resolution Orbitrap mass spectrometry (UHPLC Q-Orbitrap HRMS).

## 2. Materials and Methods

### 2.1. Chemicals and Reagents

Formic acid (FA), methanol (MeOH), and water (H_2_O) were acquired from Carlo Erba reagents (Milan, Italy). Polyphenol standards (purity > 98%) including caffeine, quinic acid, ferulic acid, *p*-coumaric acid, caffeic acid, 5-caffeoylquinic acid (5-CQA), and 3,5-dicaffeoylquinic acid (3,5-diCQA) were purchased from Sigma-Aldrich (Milan, Italy). For each standard, a stock solution at a concentration of 1 mg/mL was prepared in methanol. Working standard solutions were obtained by serial dilution and were stored at −20 °C until use.

### 2.2. Sampling

Standard Coffee (SC) and Dewaxed Coffee (DC) pods were obtained from Kimbo Caffè S.p.A. Pods are packed with a paper filter covering for use in a non-grinding espresso machine. The result is an espresso, which has a beautiful crema. Dewaxed coffee is an intense and aromatic blend, with a limited content of waxes and caffeine. The waxes of the cortical part of the grain are removed with a dewaxing process that uses an organic solvent (dichloromethane), which also extracts part of the caffeine. Thanks to the process of wax extraction, this coffee is delicate, gentle, and intense.

Roasted and ground coffee, with medium-roasted coffee beans, a blend of carefully selected fine coffees, Arabica (80%), and Robusta (20%) coffee beans from South America were chosen for both typologies of pods. Samples were stored at room temperature in their original individual packaging prior to analysis.

### 2.3. Sample Preparation

Bioactive compounds were extracted in accordance with the procedure reported by [33] with some changes. In short, 1 g of powder sample was extracted with 20 mL of mixture H_2_O:EtOH (75:25 *v*/*v*). The samples were vortexed (ZX3; VEPL Scientific, Usmate, Italy) for 1 min, sonicated (LBS 1; Zetalab srl, Padua, Italy) for 15 min, and stirred for 15 min. Then, the mixture was centrifuged for 5 min at 5000 rpm, the supernatant collected, and the pellet re-extracted another time. Finally, the two-supernatants were collected, filtrated through a 0.22 µm filter, and appropriately diluted in methanol until (1:10) further analysis.

### 2.4. UHPLC Q-Orbitrap HRMS

Polyphenolic profile was carried out by using an Ultra High-Pressure Liquid Chromatograph (UHPLC, Dionex UltiMate 3000, Thermo Fisher Scientific, Waltham, MA, USA) equipped with a Quaternary UHPLC pump working at 1250 bar, a degassing system, and an autosampler device. Chromatographic separation was performed with a thermostated (T = 25 °C) Kinetex 1.7 μm F5 (50 × 2.1 mm, Phenomenex, Torrance, CA, USA) column. The mobile phase consisted of water (A) and methanol (B) both containing 0.1% FA in. The injection volume was 1 µL. The gradient elution program was as follows: an initial 100% A, decreased to 60% A in 1 min, to 20% A in 1 min, and to 0% B in 3 min. The gradient was held for 4 min at 0% A, increased to 100% A in 2 min, and another 2 min for column re-equilibration at 100%. The total run time was 13 min. The flow rate was set at 500 µL/min. The UHPLC (Thermo Fischer Scientific, Waltham, MA, USA) system was coupled to a Q-Exactive Orbitrap mass spectrometer equipped with an electrospray (ESI) source. The mass spectrometer was operated in both positive and negative ion mode by setting a full ion MS. Full ion MS experiments were carried out with the settings: spray voltage 3.5 kV; capillary temperature 320 °C; S-lens RF level 60; sheath gas pressure 18; auxiliary gas 3; auxiliary gas heater temperature 350 °C; scan range 80–1200 *m*/*z*; microscans 1; mass resolution 35,000 full width at half maximum (FWHM); maximum injection time 200 ms; and automatic gain control (AGC) target 1 × 10^6^. For accurate mass measurement, identification was carried out at a mass tolerance of 5 ppm. Data analysis and processing were performed by using Xcalibur software, v. 3.1.66.10 (Xcalibur, Thermo Fisher Scientific, Waltham, MA, USA) [34].

### 2.5. Identification of Bioactive Compounds in Coffee Pods Samples through UHPLC-Q-Orbitrap HRMS

Identification of bioactive compounds (*n* = 14) including chlorogenic, hydroxycinnamic acids and caffeine in standard and dewaxed coffee pods was performed by using UHPLC-Q-Orbitrap HRMS analysis. The identification of bioactive compounds was carried out in Full ion MS mode. All experiments were set in negative ESI- mode, except for caffeine which showed an improved pattern in positive ESI+ mode. Satisfactory chromatography separation of analytes was achieved in a runtime of 13 min. For feruloylquinic acids (FQAs) isomers, 4-FQAand 5-FQA acids, quantification was reported as the sum because poor abundance prevents a good separation. Identification of isomers which includes CQA (*m*/*z* 353.08780), *p*-CoQA (*m*/*z* 337.09289), FCQA (*m*/*z* 367.10346), diCQA (*m*/*z* 515.11950) was carried out comparing the retention time with the standards and also by comparison of patterns previously reported in the literature [33]. Table 1 shows all the mass parameters referred to the studied compounds, such as chemical formula, theorical and measured mass (*m*/*z*), adduct ion, retention time, and accuracy.

Determination of the predominant CGAs and caffeine was carried out by using UHPLC-Q-Orbitrap HRMS analysis. Eight concentration levels were used for building the calibration curves of target analytes, and the correlation coefficients obtained were >0.99. For the semi-quantification purpose, a representative standard from the same group was used. In fact, for 3 and 5-*p*CoQA; 3, 4 and 5-FQA; 3 and 5-CQA; and 4,5-CFQA and 3,4-FCQA isomers, no standards were available. 

### 2.6. Study Design

In this single-center pilot study, a short-term nutritional intervention was performed. The four-week randomized, cross-over design was comprised of two weeks of Standard Coffee (SC) consumption and two weeks of Dewaxed Coffee (DC) consumption, separated by a two-week washout period. Randomization was performed in blocks of four using a computer-generated list, with a non-concealed allocation. 

Patients were asked to follow their habitual diet during the whole study period (Figure A1). On day one, patients received the assigned coffee (56 coffee pods of SC or DC) and were advised to consume a maximum of four pods a day. After two weeks (14 days) patients returned and received the second type of coffee (DC or SC, respectively). Adherence to coffee consumption (Number of consumed coffee pods: 0, 1, 2, 3, 4), presence of typical GERD symptoms (heartburn and regurgitation), and antacid assumption were assessed by patient entries into a tick-box diary for both study periods. Patient compliance with diary filling in was monitored and noncompliant patients were counseled. A complete clinical evaluation of Gastrointestinal Symptoms (PAGI-SYM and IBS-SSS) [35,36] and quality of life (PAGI_QoL) [37] was performed at baseline (B) and after both intervention periods at weeks two and four (SC and DC).

The PAGI-SYM (Figure A2) and PAGI-QoL (Figure A3) are standardized, and validated questionnaires were used to evaluate the severity of symptoms and the quality of life, respectively, in patients with upper gastrointestinal disorders (including GERD) over the 14 days preceding the visit [34,35,36]. Both PAGI-SYM and PAGI-QoL subscales and total rating are scored on a scale from zero (no symptoms/lowest Qol) to five (very severe symptoms/highest Qol) [38].

The IBS symptom severity scale (IBS-SSS, Figure A4) is a validated five-question survey investigating the severity of abdominal pain and distension and the dissatisfaction with bowel habits over the 10 days preceding the visit [36]. A change of 50 is adequate to detect a clinical improvement. The full versions of the questionnaires are reported in Appendix A.

Firstly, the symptom-free days (Heartburn and regurgitation) and antacid assumption over both two-week treatment periods (SC and DC) were compared. Then, the evaluation of change from the baseline PAGI-SYM (Patient Assessment of Upper Gastrointestinal Symptom Severity Questionnaire-Symptoms Severity Index), PAGI-QoL (PAGI-Quality of Life) and IBS-SSS scores were assessed. A visual description of the study design is shown in Figure 1.

### 2.7. Subjects

In total, 40 patients with a clinical and instrumental diagnosis of GERD (16 F) were recruited from the gastroenterology outpatient clinic of the University Hospital “Federico II” of Naples. The eligibility criteria were as follows: male and females aged 18–65 years, presence of typical GERD symptoms, and report of at least a one-year history of heartburn and/or regurgitation occurring at least 50% of the time following coffee consumption [39]. The exclusion criteria included: pregnancy; breastfeeding; alcohol or drug abuse; any organic gastrointestinal disease; any malignancy; any kind of organic, systemic, metabolic or autoimmune disease; history of major gastrointestinal surgery; use of proton pump inhibitor (PPI) within two weeks before screening; use of H2-blocker, prokinetics or antacids within three days before screening; or any other condition considered to be inappropriate for the study. The study was approved by the Federico II ethical committee (prot. number 70/21), and all patients gave their written consent to participate.

### 2.8. Data Analysis

A symptom-based assessment was performed to assess treatment efficacy [40,41]. The frequencies of symptom-free days (heartburn and regurgitation % from patient diary) were assessed at weeks two and four and were compared between SC and DC to verify the primary efficacy endpoint. To assess the secondary efficacy endpoint, any change of GERD-related symptoms, lower gastrointestinal symptoms and quality of life (QoL) were also evaluated using the PAGI-SYM, the IBS-SSS, and the PAGI-QoL questionnaires scores, respectively. A physical examination and an assessment of vital signs were performed at the initial appointment and at the end of both study periods.

### 2.9. Statistical Analysis

Statistical analysis was performed using Statistical Package for Social Science IBM SPSS version 25 statistical software package (Chicago, IL, USA). Continuous variables are described as mean ± standard deviation (SD), while categorical variables are described as number and frequencies. Fisher exact test, *t*-test and one-way ANOVA followed by Bonferroni post-test were used, respectively, when appropriate. All tests were two-tailed with a confidence interval of 95%. Significance was expressed at a *p*-value < 0.05.

## 3. Results

### 3.1. Quantification of Bioactive Compounds in Coffee Pods Samples through UHPLC-Q-Orbitrap HRMS

As shown in Table 2, thirteen analytes were identified, quantified, or semi-quantified in the coffee pods samples. CQAs were the most abundant investigated compounds in the coffee pods samples, with a concentration level ranging between 7.316 (DC) and 6.721 mg/g (SC). In particular, 5-CQA was the most predominant CQA in the assayed coffee pods samples, ranging from 2.928 (SC) to 3.121 (DC) mg/g powder. In the coffee pods samples investigated here, FQAs represented 5.4% (DC) to 6.6% (SC) of total CGAs with a concentration level ranging between 0.397 and 0.447 mg/g. Regarding, diCQA, concentration levels ranging between 0.107 and 0.114 mg/g represented 1.4% (DC) to 1.7% (SC) of total CGAs. Finally, *p*CoQA were found at a concentration range of 0.580 and 0.456 for DC and SC, respectively. Apart from CGAs, caffeine was quantified at a concentration level of 5.691 mg/g and 11.091 for DC and SC, respectively.

### 3.2. A randomized Pilot Study 

The demographic and baseline characteristics of all patients (*n* = 40) are summarized in Table 3.

The assessment of the percentage of symptom-free days experienced by patients during SC and DC periods showed a significant reduction of symptom frequency when consuming DC as compared to SC, with a similar number of coffees consumed during the two periods (2.7 ± 0.6 vs. 2.8 ± 0.8 for SC and DC respectively, *p* = not significant). In particular, the analysis of patient diaries proved a significant increase in both heartburn-free days and regurgitation-free days during DC compared to SC (Table 4). These findings were further supported by the observation that patients had a significant increase of antacid-free days during DC compared to SC (Table 4).

Figure 2 summarizes the individual trends for GERD-related symptoms and clearly illustrates that, after DC, a significant improvement of heartburn, regurgitation and a reduced needing of antacid assumption was reported by a majority of patients.

The overall gastrointestinal symptoms assessment showed a significant reduction in both upper and lower gastrointestinal symptoms. In particular, the total PAGI-SYM score reveals a significant improvement of upper gastrointestinal symptoms after ingestion of DC compared to SC (Table 5).

Going even more in detail, the analysis of PAGI-SYM subscales demonstrated a meaningful improvement of nausea (0.64 ± 0.64 vs. 0.29 ± 0.32; *p* < 0.01), postprandial fullness (1.69 ± 0.86 vs. 1.07 ± 0.6; *p* < 0.01), abdominal bloating (2.51 ± 1.15 vs. 1.46 ± 1.05; *p* < 0.01), upper (2.1 ± 1.19 vs. 1.09 ± 0.83; *p* < 0.01) and lower (1.36 ± 1.01 vs. 0.82 ± 0.74; *p* < 0.01) abdominal pain, heartburn, and regurgitation (1.88 ± 0.92 vs. 0.92 ± 0.56; *p* < 0.01) after the two week DC period compared to the SC period (Figure 3).

Furthermore, IBS-SSS score analysis demonstrated a significant reduction of lower gastrointestinal symptoms after DC compared to SC (Table 5). In both cases, the differences shown above were similar to those obtained when comparing the DC period with baseline conditions.

The PAGI-QoL scores analysis showed a significant improvement of quality of life after the two-week DC period compared to the SC period (Table 5). In particular, a meaningful difference in subscales for diet and food habits (1.65 ± 0.55 vs. 0.67 ± 0.39; *p* < 0.01), psychological wellbeing and distress (2.17 ± 0.58 vs. 1.1 ± 0.67; *p* < 0.01), daily activity (1.15 ± 0.61 vs. 0.75 ± 0.54; *p* < 0.01) and clothing (1.38 ± 0.25 vs. 0.5 ± 0.41; *p* < 0.05) were observed (Figure 4). Here too, the differences found comparing the DC period with basal conditions were similar to those obtained comparing the DC and SC periods.

## 4. Discussion

Until now, scarce scientific evidence for putative coffee components affecting gastric acid secretion in humans is reported in the literature [42]. Previous studies already speculated that variations in coffee type and processing might be important in the genesis of coffee-related upper gastrointestinal symptoms, yet no significant difference has emerged [39,43,44]. Although the mechanism of action is not completely understood yet, it has been hypothesized that modifying roasting conditions could reduce stomach-irritating compounds, namely caffeine, chlorogenic acids (CGAs), and N-alkanoyl-5-hydroxytryptamides (C5HTs) [45]. Caffeine is frequently investigated as the main responsible molecule in inducing GERD symptoms [46,47]. Interestingly, a recent ongoing prospective US cohort study demonstrated a minimal change in upper gastrointestinal symptoms upon stratification by caffeine status among caffeinated beverages (coffee, soda, and tea) and a major association between decaffeinated tea and GERD symptoms [48]. In line with these findings, an older experimental study demonstrated a worsening of upper gastrointestinal symptoms after caffeinated coffee consumption, but not after caffeinated tap water consumption, suggesting a feasible involvement of other unknown components of coffee in inducing GERD symptoms [49].

Overall, the results indicate that the analyzed coffee pods may represent a considerable source of CGAs and other important bioactive compounds. Concerning the CGAs in assayed coffee brew samples, the concentrations found in dewaxed coffee pods were slightly higher (*p* < 0.05) when compared to those in standard coffee pods with a concentration of 5.10 and 4.59 mg/g, respectively. According to data reported in the literature, the contents of CGAs in coffee present a large variability and are influenced by many factors, such as the variety of coffee, the roast degrees, and the brewing method used. It has been reported that, in medium roasts, a 60% loss of CGA has been observed and up to 100% loss in a dark roast. The optimal roasting condition for coffee is medium above which there is a significant reduction of bioactive compounds [50,51,52]. In general, the most studied CGAs in coffee are the three main CQA isomers, whereas diCQAs and FQAs have been barely investigated. Several investigations have reported the capability of CGAs to positively modulate important biological status, maintain health, and exert a pivotal role in the reduction of risk of a variety of diseases [53,54]. Our findings confirm that coffee pods even after the dewaxed process, maintain a considerable source of CGAs and other important bioactive compounds correlated with the reduction of risk of a variety of diseases.

Patients with GERD often implicate coffee in causing or worsening reflux symptoms such as heartburn and regurgitation, thus leading to coffee avoidance [31,32,39,42]. Furthermore, the lack of defined and standardized guidelines leads physicians to frequently recommend limiting coffee consumption in patients with GERD. Previous research has already tried to identify an existing coffee type or product that is less likely to trigger typical reflux symptoms in coffee-sensitive individuals, without any significant results [55]. Dewaxing is an innovative procedure in which the waxy layer is removed from unroasted coffee together with a small amount of caffeine with an organic solvent. Although our study is limited by the absence of a caffeine controlled interventional arm, we believe that our main findings showed that, in a large well-selected population of coffee-sensitive patients with GERD, chronic DC consumption:

(1). Was associated with an increase of symptom-free days and antacid-free days compared to SC; 

(2). Led to a reduction of both upper (PAGI-SYM) and lower (IBS-SSS) gastrointestinal symptoms compared to SC; 

(3). Improved gastrointestinal-related quality of life (PAGI-QoL) compared to SC. While still preliminary, data obtained from the present pilot study provide promising evidence for the efficacy of DC consumption in patients with GERD. Particularly, DC seems to be better tolerated, does not compromise the quality of life, and does not affect gastrointestinal well-being in coffee-sensitive patients with GERD.

## 5. Conclusions

From the randomized pilot study emerged evidence that dewaxed coffee pod consumption give a significant reduction in both upper and lower gastrointestinal symptoms frequency. The analysis of PAGI-SYM subscales demonstrated a meaningful improvement of nausea, postprandial fullness, abdominal bloating, upper and lower abdominal pain, heartburn, and regurgitation after the two-week DC period compared to the SC period, ameliorating the quality of life in patients with functional gastrointestinal symptoms.

Wider and longer randomized trials are needed to confirm our results and to better understand the link between waxes, caffeine content, and gastroesophageal symptoms. Furthermore, an eventual confirmation of our findings could be extremely useful to limit dietary restrictions often suggested to patients with GERD. Although our findings need to be further studied and are far from being considered as a treatment option for patients suffering from gastroesophageal symptoms, we believe that, being well tolerated, DC can be considered an option to reintroduce coffee consumption in patients with GERD.

## Figures and Tables

**Figure 1 nutrients-14-02510-f001:**
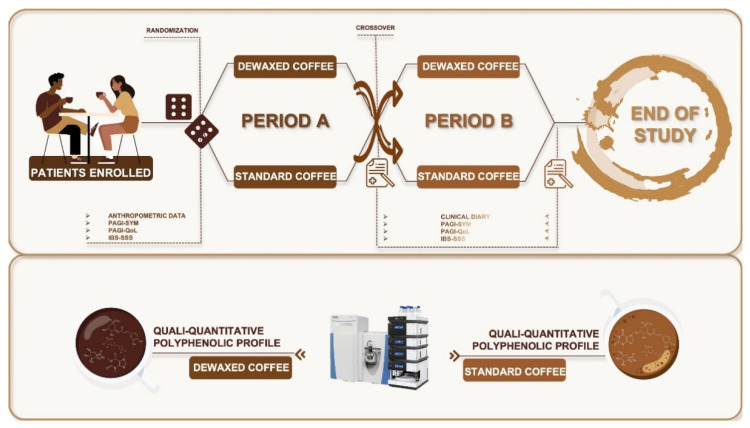
Visual description of study design. IBS-SSS, Irritable Bowel Syndrome—Symptom Severity Scale; PAGI-QoL, Patient Assessment of Upper Gastrointestinal Disorders-Quality of Life; PAGI-SYM, PAGI-Symptoms Severity Index.

**Figure 2 nutrients-14-02510-f002:**
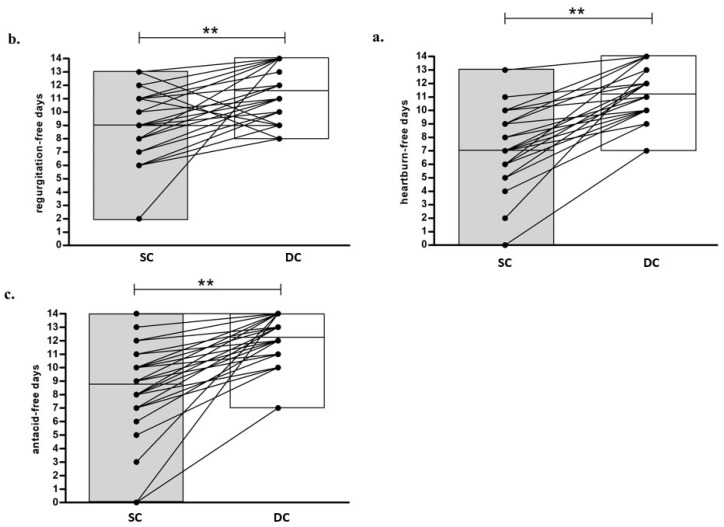
Evaluation of heartburn-free days (**a**), regurgitation-free days (**b**) and antacid-free days (**c**) during both treatment periods (*n* = 40). DC, Dewaxed Coffee; SC, Standard Coffee. ** *p* < 0.01.

**Figure 3 nutrients-14-02510-f003:**
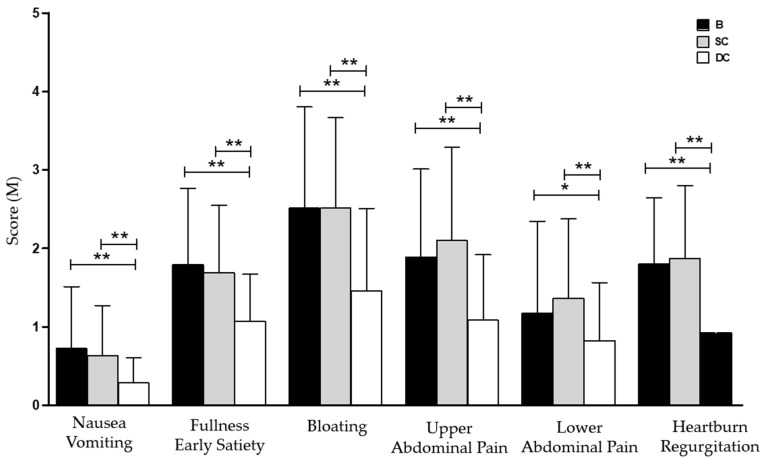
PAGI-SYM subscales score at basal condition and after both treatment periods. Data are presented as mean ± SD (*n* = 40). B, Basal Conditions; DC, Dewaxed Coffee; SC, Standard Coffee. * *p* < 0.05; ** *p* < 0.01.

**Figure 4 nutrients-14-02510-f004:**
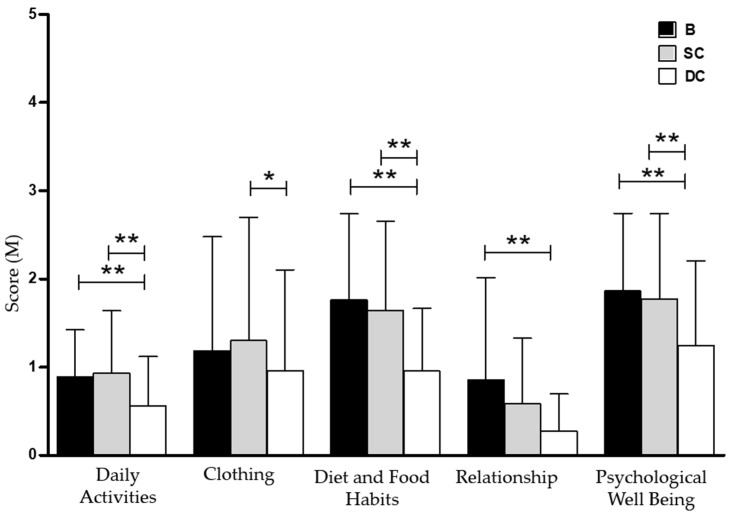
PAGI-QoL subscales score at basal condition and after both treatment periods. Data are presented as mean ± SD (*n* = 40). B, Basal Conditions; DC, Dewaxed Coffee; SC, Standard Coffee. * *p* < 0.05; ** *p* < 0.01.

**Table 1 nutrients-14-02510-t001:** UHPLC-MS parameters of the assayed analytes (*n* = 14).

Compound *	Chemical	Adduct	RT *	Measured	Theoretical	Accuracy
	Formula	Ion	(min)	Mass (*m*/*z*)	Mass (*m*/*z*)	(Δ mg/kg)
Quinic Acid	C_7_H_12_O_6_	[M−H]^−^	1.12	191.05531	191.05611	4.18
3-*p*CoQA	C_16_H_18_O_8_	[M−H]^−^	2.84	337.09232	337.09289	−1.69
3-FQA	C_17_H_20_O_9_	[M−H]^−^	3.03	367.10367	367.10346	−0.57
Caffeic Acid	C_9_H_8_O_4_	[M−H]^−^	3.07	179.03426	179.03498	4.02
5-CQA	C_16_H_18_O_9_	[M−H]^−^	3.09	353.08813	353.08780	−0.93
4-CQA	C_16_H_18_O_9_	[M−H]^−^	3.10	353.08901	353.08780	−3.42
3-CQA	C_16_H_18_O_9_	[M−H]^−^	3.12	353.08852	353.08780	−2.03
Caffeine	C_8_H_10_N_4_O_2_	[M + H]^+^	3.20	195.08751	195.08765	0.72
*p*-Coumaric acid	C_9_H_8_O_3_	[M−H]^−^	3.25	163.03926	163.04006	4.91
5-*p*CoQA	C_16_H_18_O_8_	[M−H]^−^	3.27	337.09389	337.09289	−2.97
3,4-diCQA	C_25_H_24_O_12_	[M−H]^−^	3.28	515.12036	515.11950	−1.67
4 + 5-FQA	C_17_H_20_O_9_	[M−H]^−^	3.34	367.10303	367.10346	4.72
Ferulic Acid	C_10_H_10_O_4_	[M−H]^−^	3.38	193.05017	193.05063	−2.38
3,5-diCQA	C_25_H_24_O_12_	[M−H]^−^	3.45	515.12036	515.11950	−1.67

* Abbreviations: CQA: Caffeoylquinic; *p*CoQA: *p*-Coumaroylquinic acid; FQA: Feruloylquinic acid; diCQA: Dicaffeoylquinic acid, RT: retention time.

**Table 2 nutrients-14-02510-t002:** Chlorogenic acids and other bioactive compounds (*n* = 14) content in standard and dewaxed coffee pods samples. Data are displayed as average value (mg/g) and standard deviation (±SD).

Compound *	Dewaxed Coffee		Standard Coffee	
	mg/g	±SD	mg/g	±SD
Quinic Acid	0.672	0.049	0.684	0.033
3-*p*CoQA	0.509 ^a^	0.001	0.404 ^b^	0.002
3-FQA	0.094	0.003	0.103	0.005
Caffeic Acid	0.022 ^a^	0.001	0.015 ^b^	0.002
5-CQA	3.132 ^a^	0.016	2.928 ^b^	0.017
4-CQA	1.034 ^a^	0.012	0.928 ^b^	0.013
3-CQA	0.932 ^a^	0.005	0.728 ^b^	0.013
Caffeine	5.691 ^a^	0.07	11.091 ^b^	0.11
*p*-Coumaric acid	NF		NF *	
5-*p*CoQA	0.071 ^a^	0.007	0.053 ^b^	0.003
3,4-diCQA	0.083	0.001	0.086	0.002
4 + 5-FQA	0.303 ^a^	0.024	0.344 ^b^	0.008
Ferulic Acid	0.440 ^a^	0.094	0.420 ^b^	0.033
3,5-diCQA	0.025 ^a^	0.000	0.028 ^b^	0.001

* Abbreviations: CQA: Caffeoylquinic; *p*CoQA: *p*-Coumaroylquinic acid; FQA: Feruloylquinic acid; diCQA: Dicaffeoylquinic acid; SC: standard coffee pods; DC: dewaxed coffee pods, NF: Not found. Tukey’s test was used to evaluate differences between SC and DC samples considering *p*-value less than 0.05 as significant. ^a, b^ Different letters show a significant difference (*p* < 0.05) between SC and DC samples.

**Table 3 nutrients-14-02510-t003:** Demographic and baseline characteristics of patients. Values are means ± SD unless otherwise indicated; *n* = 40 patients.

Patients	T0
Age (years)	41.5 ± 12
Sex *n* (%)	F 16 (40)
Weight (kg)	75.3 ± 15.9
Height (m)	1.7 ± 0.1
BMI (kg/m^2^)	25.5 ± 4
Smoke *n* (%)	13 (32.5)
Physical Activity *n* (%)	19 (47.5)

**Table 4 nutrients-14-02510-t004:** Symptoms and Antacid-Free Days during SC and DC Treatment Periods. Data are presented as percentages (%).

	SC	DC	*p*-Value
Heartburn-free days, %	50.18 ± 17.46	79.82 ± 10.84	*p* < 0.05
Regurgitation-free days, %	64.46 ± 14.87	82.68 ± 12.83	*p* < 0.05
Antacid-free days, %	62.5 ± 22.22	87.5 ± 11.29	*p* < 0.05

Abbreviations: DC: Dewaxed Coffee; SC: Standard Coffee.

**Table 5 nutrients-14-02510-t005:** PAGI-SYM, PAGI-QoL and IBS-SSS total scores changes in basal conditions and during SC and DC Treatment Periods. Data are presented as mean ± SD.

	B	SC *	DC *	*p*-Value *
PAGI-SYM	1.6 ± 0.75	1.7 ± 0.72	0.9 ± 0.48	*p* < 0.01
PAGI-QoL	1.3 ± 0.73	1.2 ± 0.81	0.8 ± 0.64	*p* < 0.01
IBS-SSS	196.9 ± 71.61	215.65 ± 68.51	149.75 ± 56.97	*p* < 0.01

B, basal conditions; DC, Dewaxed Coffee; IBS-SSS, Irritable Bowel Syndrome—Symptom Severity Scale; PAGI-QoL, Patient Assessment of Upper Gastrointestinal Disorders-Quality of Life; PAGI-SYM, PAGI-Symptoms Severity Index; SC, Standard Coffee. * (DC vs. SC).

## Data Availability

All data generated or analyzed during this study are included in this article. Further inquiries can be directed to the corresponding author.

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
