# Peer review of "Effect of Dewaxed Coffee on Gastroesophageal Symptoms in Patients with GERD: A Randomized Pilot Study"

_nutrients, 2022, doi:10.3390/nu14122510_

Round 1
Reviewer 1 Report
The study addressed a useful topic and got positive results that appeared fairly consistent trend wise among the subjects. Unfortunately, the study design and presentation had serious problems. The biggest problems are as follows.
It does not appear that the protocol was approved by an IRB. If that is true, this by itself warrant that the manuscript not be published.
The manuscript did not indicate how much coffee the participants used prior to the study. More importantly, the study did not control the timing of coffee intake nor much coffee per day was taken except to set a maximum. It’s possible that the subjects consumed more of the regular coffee because they liked it better. That would affect the results.
Were all the effects seen simply due to different amounts of caffeine? If yes, that the dewaxed coffee is not overly special for reducing heartburn actions.
In addition to the design problems, the following presentation issues occurred. These can be fixed, but that wouldn’t overcome the design problems.
The authors should have mentioned that they got an improvement, but not a cure.
It was never clear why the polyphenol analysis was done. In the Discussion, it was implied that the polyphenols could contribute to heartburn, though some past research suggests protection potential. In the conclusion, I finally got the idea that maybe the polyphenols were analyzed just to say that the 2 coffees were similar in potential health benefits.
What polyphenol classes were being measured wasn’t clear. The text talks about chlorogenic acids, but Table 2 never uses that term. Are all the categories a type of chlorogenic acid? Also, the abstract uses 2 different abbreviations for chlororogenic acids.
The authors did not justify a lack of a washout period.
Section 3.1 belongs in the Methods, not the Results.
In Table 3, no units nor definition is given for the physical activity measurement.
The Conclusion runs too long
Author Response
Manuscript ID: nutrients-1737415
Title: Effect of dewaxed coffee on gastroesophageal symptoms in patients with GERD: a randomized pilot study Response to Reviewer 1
The study addressed a useful topic and got positive results that appeared fairly consistent trend wise among the subjects. Unfortunately, the study design and presentation had serious problems. The biggest problems are as follows.
Point 1: It does not appear that the protocol was approved by an IRB. If that is true, this by itself warrant that the manuscript not be published.
Response 1: We thank the reviewer for this comment. We must apologize for this issue that was due to our error in the file upload. The study was approved by the local ethical committee. We stated this point in the subject’s section of the revised version of the manuscript.
Point 2: The manuscript did not indicate how much coffee the participants used prior to the study. More importantly, the study did not control the timing of coffee intake nor much coffee per day was taken except to set a maximum. It’s possible that the subjects consumed more of the regular coffee because they liked it better. That would affect the results.
Response 2: We agree with the reviewer’s comment. We set a preliminary analysis of the studied population by considering patients usual habits in coffee consumption (subjects referred to consume 2-4 cups of coffee per day). For this reason, we decided to set 4 cups per day as maximum during the study protocol. We thank the reviewer for the very helpful comment about the number of coffees per day in the two treatment groups. We checked patients diary and verified that there were no difference in palatability, nor number of coffees per day. This point has been added in the revised version of the manuscript, ruling out the possibility to a putative bias.
Point 3: Were all the effects seen simply due to different amounts of caffeine? If yes, that the dewaxed coffee is not overly special for reducing heartburn actions.
Response 3: The registered effects regard the relationship between standard and dewaxed coffee intake on GERD. Dewaxed coffee contains about 50% less caffeine, but this is not the most important point. From the comparison emerges that dewaxed coffee ameliorates the GERD-like symptoms. Basically, the effect can be subject to the dewaxed process, a mild innovative treatment of extraction in which the waxy layer is removed from unroasted coffee together with a small amount of caffeine with an organic solvent. The waxy elements are naturally present in the cortical part of the coffee bean. The solubility properties of waxes make them barely digestible and difficult to absorb for the human body. Furthermore, they could be able to induce a mild irritation of gastric mucosa in predisposed subjects.
Point 4: In addition to the design problems, the following presentation issues occurred. These can be fixed, but that wouldn’t overcome the design problems. The authors should have mentioned that they got an improvement, but not a cure.
Response 4: Again, we thank the reviewer for the helpful comment. We absolutely agree with this point, the authors do not want to promote a cure for GERD symptoms, but only a viable alternative to make coffee more
better tolerated that enables to reintroduce of coffee consumption to the nutritional habits of patients with GERD.
Point 5: It was never clear why the polyphenol analysis was done. In the Discussion, it was implied that the
polyphenols could contribute to heartburn, though some past research suggests protection potential. In the
conclusion, I finally got the idea that maybe the polyphenols were analyzed just to say that the 2 coffees were
similar in potential health benefits.
Response 5: A chemical profile of coffee pods was performed by UHPLC-Q-Orbitrap spectrometry measurement, CGAs and caffeine were quantified. Results of the herein scientific study confirm that coffee pods even after the dewaxed process maintain a considerable source of CGAs and other important bioactive
compounds correlated with the reduction of risk of a variety of diseases. As suggested by reviewer 1, the authors moved this part to the discussion section.
Point 6: What polyphenol classes were being measured wasn’t clear. The text talks about chlorogenic acids, but Table 2 never uses that term. Are all the categories a type of chlorogenic acid? Also, the abstract uses 2
different abbreviations for chlororogenic acids.
Response 6: Table 2 does not refer only to chlorogenic acids as more bioactive compounds are analyzed, which incorporate the chlorogenic acids present in greater quantities. As suggested by reviewer 1, the authors clarify this part and change the abbreviations for chlorogenic acids in the abstract section.
Point 7: The authors did not justify a lack of a washout period.
Response 7: As suggested by reviewer 1, we added this point in the revised manuscript.
Point 8: Section 3.1 belongs in the Methods, not the Results.
Response 8: As suggested by reviewer 1, we moved part of this section to the methods.
Point 9: In Table 3, no units nor definition is given for the physical activity measurement.
Response 9: As rightly suggested by reviewer 1, the authors added this missing information.
Point 10: The Conclusion runs too long
Response 10: As suggested by reviewer 1, the authors reduced the conclusions section.
The authors thank reviewer 1 for evaluating our manuscript.

Reviewer 2 Report
Figure 2 a,b,c. Please change CS and CD to DC in DC
Author Response
Manuscript ID: nutrients-1737415
Title: Effect of dewaxed coffee on gastroesophageal symptoms in patients with GERD: a randomized pilot study
Response to Reviewer 2 Point 1: Figure 2 a,b,c. Please change CS and CD to DC in DC
Response 1: As suggested by reviewer 2, the author change CS and CD to SC in DC.
The authors thank reviewer 2 for evaluating our manuscript.

Reviewer 3 Report
The nutrients-1737415 is an interesting topic since coffee consumption is rapidly growing and there are pros and cons to taking coffee. GERD is a disease that may become severe with excess caffeine consumption (i.e. coffee). However, it is very interesting research authors should update their manuscript.
1. There are multiple varieties of coffee beans, roasting, and brewing methods. However, the authors did not mention any of these in the manuscript; therefore, the authors should provide detailed information about coffee.
2. Also, authors should acknowledge varieties of coffee beans, roasting, and brewing methods may alter components in coffee (i.e. caffeine, chlorogenic acid, and so on) with decent references.
3. Please provide the definition and fabrication methods for the dewaxed coffee.
4. GERD is a disease; however, the authors only relied on the survey without a doctor’s diagnosis. Therefore, authors should not use GERD in the manuscript.
5. IACUC approval is required unless the nutrients-1737415 are not acceptable due to ethical issues.
6. [L 29,187] Spacing issues.
7. [Table 2] Statistical identification is required.
8. [Figure 2] Re-check the groups. (CS, CD vs. SC, DC)
9. [L298] (A) Reference(s) is/are required.
10. [L310] Statistical difference should be acknowledged (5.10 vs 4.59 mg/g).
11. Please provide detailed information for the DC and SC.
12. In the discussion, the authors should describe what is the main reason to reduce GERD-like symptoms by consumption of dewaxed coffee (caffeine, chlorogenic acid, and/or others).
Author Response
Manuscript ID: nutrients-1737415
Title: Effect of dewaxed coffee on gastroesophageal symptoms in patients with GERD: a randomized pilot study
Response to Reviewer 3
The nutrients-1737415 is an interesting topic since coffee consumption is rapidly growing and there are pros and cons to taking coffee. GERD is a disease that may become severe with excess caffeine consumption (i.e. coffee). However, it is very interesting research authors should update their manuscript.
Point 1: There are multiple varieties of coffee beans, roasting, and brewing methods. However, the authors did not mention any of these in the manuscript; therefore, the authors should provide detailed information about coffee.
Response 1: As suggested by reviewer 3, the authors added the missing information in the manuscript.
Point 2: Also, authors should acknowledge varieties of coffee beans, roasting, and brewing methods may alter components in coffee (i.e. caffeine, chlorogenic acid, and so on) with decent references.
Response 2: As rightly suggested by reviewer 3, the authors added the missing information in the manuscript.
Point 3: Please provide the definition and fabrication methods for the dewaxed coffee.
Response 3: As suggested by reviewer 3, the authors added the missing information in the manuscript.
Point 4: GERD is a disease; however, the authors only relied on the survey without a doctor’s diagnosis. Therefore, authors should not use GERD in the manuscript.
Response 4: We absolutely agree with this reviewer. GERD is a clinically based diagnosis, and this is our case. The patients were indeed selected according to internationally accepted clinical criteria. We better stated this point in the revised version of the manuscript.
Point 5: IACUC approval is required unless the nutrients-1737415 are not acceptable due to ethical issues.
Response 5: We thank the reviewer for this comment. We must apologize for this issue that was due to our error in the file upload. The study was approved by the local ethical committee. We stated this point in the subject’s section of the revised version of the manuscript.
Point 6: [L 29,187] Spacing issues.
Response 6: As suggested by reviewer 3, the authors checked the spacing issues.
Point 7: [Table 2] Statistical identification is required.
Response 7: As suggested by reviewer 3, the authors added the statistics in Table 2.
Point 8: [Figure 2] Re-check the groups. (CS, CD vs. SC, DC)
Response 8: As suggested by reviewer 3, the authors checked the groups in Figure 2.
Point 9: [L298] (A) Reference(s) is/are required.
Response 9: As suggested by reviewer 3, the authors added an appropriate reference in line 298.
Point 10: [L310] Statistical difference should be acknowledged (5.10 vs 4.59 mg/g).
Response 10: As suggested by reviewer 3, the authors added the significant difference.
Point 11: Please provide detailed information for the DC and SC.
Response 11: As suggested by reviewer 3, the authors provided further details for DC and SC typologies.
Point 12: In the discussion, the authors should describe what is the main reason to reduce GERD-like symptoms by consumption of dewaxed coffee (caffeine, chlorogenic acid, and/or others).
Response 12: We thank the reviewer for the comment. We cannot conclude what of the component is beneficial for this subset of patients, for this reason we stated this point in the revised manuscript. We took the reviewer point and hopefully we will be able to address this interesting issue in a next study.
The authors thank reviewer 3 for evaluating our manuscript.

Round 2
Reviewer 1 Report
The submission was improved substantially by adding various pieces of information. I do still have few important, by easily addressed comments on the writing:
1. The name of the IRB board should be stated.
2. On Ln 163, I found the sentence confusing. Was a washout used? If not, why not? If not using a washout was based on a previous study, the study should be cited or listed as unpublished results.
3. The beginning part of the conclusion should use past tense verbs.
4. The Discussion and conclusions should note that symptoms were improved, not eliminated
Author Response
Manuscript ID: nutrients-1737415
Title: Effect of dewaxed coffee on gastroesophageal symptoms in patients with GERD: a randomized pilot study
Response to Reviewer 1
The submission was improved substantially by adding various pieces of information. I do still have few important, by easily addressed comments on the writing:
Point 1: The name of the IRB board should be stated.
Response 1: The study was approved by the Federico II ethical committee.
Point 2: On Ln 163, I found the sentence confusing. Was a washout used? If not, why not? If not using a washout was based on a previous study, the study should be cited or listed as unpublished results.
Response 2: As indicated in Line 163, the study design involved washout phases. As suggested by the reviewer, the authors modified the sentence.
Point 3: The beginning part of the conclusion should use past tense verbs.
Response 3: As suggested by the reviewer, the authors modified the sentence.
Point 4: The Discussion and conclusions should note that symptoms were improved, not eliminated
Response 4: As suggested by the reviewer, the authors modified some sentences in the discussion and conclusions section. We absolutely agree with this point, the authors do not want to promote a cure for GERD symptoms, but only a viable alternative to make coffee better tolerated that enables to reintroduce of coffee consumption to the nutritional habits of patients with GERD.
The authors thank reviewer 1 for the careful evaluation of our manuscript..
